# Learning about Grandparents’ Past Life: Reflections of Grandchildren in an Intergenerational Reminiscence Project for Asian American Families

**DOI:** 10.3390/bs13090733

**Published:** 2023-09-01

**Authors:** Ling Xu, Aaron Hagedorn, Minjaal Raval

**Affiliations:** School of Social Work, University of Texas at Arlington, Arlington, TX 76010, USA; aaron.hagedorn@uta.edu (A.H.); mmr8972@mavs.uta.edu (M.R.)

**Keywords:** Asian American families, grandparent–grandchild, life review, intergenerational, reminiscence

## Abstract

Immigration, aging, and dementia often result in a triple jeopardy for Asian American older adults. To improve the well-being of Asian American older adults as well as generational bonding, an Intergenerational Grandparent–Grandchild Reminiscence Program was developed. This paper qualitatively reports on the weekly reflections from the grandchild participants of this program. Older grandparents received six sessions of life-review discussion with their grandchildren remotely or in person for approximately 1 h each week for 6 weeks. Each grandchild (*n* = 12) provided a written reflection each week after talking with their grandparent. The qualitative data were organized and analyzed using the five phases of the rigorous and accelerated data reduction (RADaR) technique. The results show three categories of themes, as follows: Category 1—Positive experience: more connection with the grandparent; learning more about the grandparent’s past life experience; and more engagement; Category 2—Challenging experience: over-explaining things; language or vocabulary barriers; and overly-broad topics; Category 3—Strategy to lead the discussion: using guiding questions in the manual; using translators; spending time together; and taking notes. The results show that the intergenerational reminiscence program is promising for bonding and connection in the grandparent–grandchild relationship. Participants also gained knowledge and experienced challenges when talking with their grandparents during the program.

## 1. Introduction

Asian Americans are the fastest-growing ethnic group in the U.S. There are approximately 24.0 million individuals who identify as Asian alone or in combination in the United States in 2020, as reported in the 2020 census [1]. Many in this population immigrated to the U.S. at older ages [2,3]. Older immigrants face the greatest mental health problems among all the immigrant populations, especially those who arrive later in life [4,5], and thus, experience additional challenges, such as limited acculturation and language ability, lower socioeconomic status, ineligibility for entitlement and welfare programs, and isolation [6,7]. Asian American older adults also face numerous challenges, including physical, psychological, and mental health issues [8,9,10], as well as racial discrimination on a regular basis [11], especially after COVID-19 [12,13]. As the U.S. population becomes older and more diverse, programs and interventions on how to improve the mental health of older immigrants are sorely needed.

Another common health issue seen among older adults is dementia. Immigration, aging, and dementia often result in a triple jeopardy for Asian Americans. Immigrating to a new country is a transformative experience that frequently brings about heightened stress and challenges, as mentioned above [6,7,8,9,10,11]. Moreover, the process of aging introduces an extra layer of difficulty to the immigration journey. A triple jeopardy emerges when older individuals also face additional adversities such as dementia. It is commonly seen among people who are diagnosed with dementia that they start experiencing preoccupation with their diagnosis, hypervigilance, precipitating a crisis, anxiety, and negative effects on their self-esteem [14]. The combination of this triple jeopardy has a profound impact on Asian American older adults, potentially resulting in traumatic experiences and feelings of shame regarding their thoughts. Dealing with these challenges may necessitate substantial emotional support for them. Therefore, it is imperative to urgently explore strategies that can effectively promote and safeguard their resilience to emotional well-being. However, few studies have addressed issues in developing, implementing, and testing intervention strategies to promote emotional well-being among older immigrants who have cognitive impairment.

### 1.1. Intergenerational Reminiscence Approach

The use of intergenerational connections is helpful in promoting the well-being of older adults in general. It is reported that familial, intergenerational bonds and relationships play a major role in Asian American families, which also emphasize intergenerational solidarity between grandparents and grandchildren [15]. However, due to language barriers, acculturation, and different educational backgrounds, etc., the intergenerational bond between grandparents and grandchildren has been threatened [16,17].

The reminiscence approach has been used in the literature to help older adults. The process of reminiscence usually consists of discussing past life events through experiences facilitated by the use of familiar items and objects, such as photographs and music [18]. It has shown promising results for interventions in relieving depression [19,20,21] and improving the social and emotional well-being of older adults [22]. The reminiscence intervention/process has also been shown to have positive effects on people with dementia and is seen as a psychological intervention with a strong and reliable evidence base [23,24].

The research suggests that reminiscence combined with an intergenerational approach may yield significant social and mental health benefits for older adults. Through intergenerational reminiscence, older persons may pass on their life experiences and life lessons to younger generations [25,26], and the sharing of autobiographical memories has both social and psychological functions [27]. Previous studies also assert that the use of an intergenerational approach is an effective strategy for promoting positive social and mental health in later life, such as reducing social isolation and loneliness [28]. Intergenerational contact and relationships between older people and young adults can prevent stereotypes or negative attitudes from forming or refute existing ones [29]. Additionally, the outcomes of intergenerational programming include the younger generation’s ability to build relationships with older adults and positive changes in their attitudes toward older adults [30]. Moreover, using the younger generation as volunteers rather than professionals offers a more sustainable and cost-effective intervention for implementation in community-based settings. However, as far as the authors are aware, there have been limited studies that utilize grandchildren as a young generation or volunteers to enhance the well-being of older adults.

### 1.2. The Present Study

The above-mentioned research evidence provides a foundation for the design and development of the Intergenerational Grandparent–Grandchild Reminiscence Program. In this program, Asian American grandparents who self-reported having early-stage dementia were paired with one of their grandchildren (aged 18 years and above). They participated in weekly discussions for approximately 1 h, where the focus was on life review and reminiscence of the grandparents, facilitated by their grandchildren. The objectives of this research project were to use intergenerational grandparent–grandchild reminiscence conversation to help promote the emotional well-being of older adults and intergenerational bonding in Asian American families. In the present study, we delved into and documented the experiences and feedback of the grandchildren following each week’s conversations by analyzing the reflection pages they submitted each week.

## 2. Materials and Methods

### 2.1. Process of Data Collection

Throughout the entire research project, an exploratory mixed method design was used to integrate both quantitative (survey at pre- and post-tests) and qualitative (reflection pages each week and interview after the study) data collections. The research project was approved by the Institutional Review Board of the authors’ institution (IRB# 2020-0210). Data were collected from January 2021 to December 2022.

After the participants (grandparent–grandchild dyads) signed the informed consent, the grandchildren participants were offered a manual of calling instructions and participated in a 1 h mandatory online training. Only after the grandchildren completed the training could they start the weekly discussions with their grandparent partner. Then, grandparents received 6 sessions of life review reminiscence with their grandchildren in person or virtually (approximately 1 h each week for 6 weeks). The process was a series of discussions using themes of major moments in their life. The themes of the reminiscence-based conversation included: (1) major turning points in life, (2) family history, (3) life/career accomplishments, (4) history of loves and hates (likes and dislikes), (5) experiences of suffering or stressful experiences, and (6) meaning and purpose of life. A manual of interaction guidelines was prepared for the grandchildren to lead the interactions with their grandparents based on the intervention manual proposed by Watt and Cappeliez [31,32], which outlined the implementation of integrative and instrumental reminiscence interventions that promote acceptance of self, conflict resolution and reconciliation, a sense of meaning and self-worth, recalling how one coped with past problems, and drawing from past experience to solve the present problems.

Following each week’s conversation, each grandchild participant submitted a reflection on their experiences, resulting in a total of six reflections for each participant. Some participants went into great detail and exceeded one page in their reflection submissions. The participants submitted the weekly reflections to the pre-created folder in OneDrive, a university-approved secure platform. Only the research team members and the grandchild had access to their specific folders. For this particular paper, our focus lies solely on presenting the qualitative aspect of the grandchildren’s weekly reflections.

### 2.2. Samples

The grandchild participants recruited in this study were adults aged 18 or above, self-identified as Asian American from 4 ethnic groups (Chinese, Korean, Vietnamese, Indian), had a grandparent (aged 65 or above) with a recent dementia diagnosis in the past 12 months (self-reported), and could speak the same language as their grandparent. The majority of the grandchild participants were recruited from the Dallas–Fort Worth (DFW) area.

Twelve grandchildren completed submitting their weekly reflection pages. The demographic information is reported in Table 1. As shown in Table 1, the participants had an average age of 21.17 (*SD* = 3.01, *Range* = 18–26). Around two-thirds of them were granddaughters (66.7%). Half of them (*n* = 6) were still in college, and one-fourth of them had obtained a bachelor’s degree (*n* = 3). The majority of them were Chinese American (*n* = 9, 75%). Except for one participant who immigrated to the U.S. in childhood, all other participants were born in the United States. The majority of the grandchildren lived in the same state as their grandparent (*n* = 5, 41.7%) or in another state within the USA (*n* = 5, 41.7%). More than half of them were aware of their grandparents’ memory issues (*n* = 7, 58.3%). All of the participants reported their health as “good” (*n* = 7, 58.3%), “very good” (*n* = 3, 25%), or “excellent (*n* = 2, 16.7%).

### 2.3. Instrument

After the conversation with their grandparents each week, the grandchildren were required to fill out a structured reflection page that asked about their experiences and feelings towards that week’s conversation. Four general reflection questions were asked each week for each grandchild, which included:How was this experience? How did you feel during the conversations?What did you feel was the most helpful/beneficial during this session?What did you feel was the least helpful/beneficial during this session?Other reflections/takeaways?

### 2.4. Data Analysis

For the analysis of the reflection pages in the present study, the five phases of the rigorous and accelerated data reduction (RADaR) technique were used [33]. The first step of the analysis was to organize and format all the data (transcripts) in a uniform manner. This was achieved by putting all the data into Microsoft Excel spreadsheets in an all-inclusive table. The data were put into different columns and rows with different headings, such as the transcript number, participant numbers, reflection questions, participants’ responses, codes, themes, and notes as a part of *Phase 1.* Then, as a part of *Phase 2*, the researchers analyzed and removed the data that were not relevant to the research question; thus, in *Phase 2*, only the data that were relevant in answering the research question remained. Three members of the research team then coded the data, individually and then jointly, to ensure that the data that remained were relevant and needed to answer the research question. All the rows and columns that were not needed were deleted. The data in *Phase 2* were then further analyzed by highlighting the relevant data and coding it to obtain more reduced data for further coding in *Phase 3*, which focused more on “open codes”. In *Phase 3*, the researchers moved from open codes to more focused codes [33,34] which only included the highlighted data from the previous table that were the most relevant to the research question, leading to *Phase 4*. In *Phase 4*, the researchers then analyzed the focused codes and interpreted the data. In the final phase, *Phase 5*, the researchers organized the interpreted data into the emerging themes and project deliverables.

## 3. Findings/Results

The weekly reflections from the grandchild participants were reported and analyzed in the present study. The findings show the emergence of three categories with themes. Category 1: Positive experience. The grandchildren expressed: (1) more connection with their grandparent, (2) learned more about their grandparent’s past life experiences, and (3) had more engagement. Category 2: Challenging experience. The grandchildren felt challenges in: (1) over-explaining things, (2) language or vocabulary barriers, (3) overly-broad topics. Category 3: Strategy to lead the discussion. The grandchildren reported the following good ways to facilitate their weekly conversation with grandparents: (1) using guiding questions in the manual; (2) using translators, (3) spending time together, and (4) taking notes.

### 3.1. Category 1: Positive Experience

Overall, the participants had a positive and insightful experience from this study. The following themes emerged when their experiences were analyzed.

#### 3.1.1. More Connection with Grandparent

Many participants expressed how they felt more connected and a closer bond with their grandparents after weekly calls with them. For example, one of the participants said:

“It was a very valuable experience. I felt a lot more connected to my grandfather during this conversation because it gave me the opportunity to learn and ask questions I don’t usually consider when I spend time with him.”(Week 1, #4)

“Good! I learned a lot about my grandfather’s hardships, and also was able to connect with my grandfather and get advice from him. I really felt I was able to connect with my grandfather on a deeper level.”(Week 5, #4)

Having had a conversation about their grandparent’s life review, another participant also talked about how they gained more understanding of the lives of their grandparents as well as their parents, and thus, felt more connections with both generations.

“…I knew that a lot of challenges that my grandmother faced with my mother involved me and how to raise me. I do feel that it was also a major learning experience because this was not something that I was fully aware of as a child or that many of the hardships were hidden from me to give me a better childhood… I now felt more emotional and psychologically connected and understanding of their lives”(Week 5, #11)

While for most of the participants the experience was positive, one participant expressed having felt a negative experience with some topics, such as the topic of stressful life situations. For example, participant #5 shared: “It was nice to hear how my grandpa felt about the whole experience from his own words. Didn’t love the topic since it was a bit more somber and not pleasant, but that’s ok.” (Week 5, #5)

#### 3.1.2. Learn More about Grandparent’s Past Life Experience

The participants reported how beneficial this experience was for them in learning more about their grandparents, their past life, struggles, and experiences. For example, one participant expressed:

“This experience allowed me to understand more about her [grandma]. She was telling me about her experience working in the fields and gardening. This reminds me of the time that she would often garden at my house. This makes me understand why so would always garden because she has had the experience. Although she has mentioned that it was hard work in Vietnam, it makes me wonder why she would continue to do so after she moved to the states. Nowadays, she mentioned that she does not garden and she’s okay with that as she enjoys just resting at home. I also asked her what her dreams were and what her children (my aunts and uncles) did for work. She did not give exact answers, most likely because she forgot, but she would often refer to how the Vietnam War affected her. As a result, it can be seen that the Vietnam War played a big part in creating career challenges for her and our family.”(Week 3, #6)

Another participant talked about how she got to know her mother’s childhood and her grandparent’s different family relationships. The participant stated:

“I think this reflection had given me more insight regarding the upbringing of my mother and the various family relationships that my grandparent had at that time. I felt the same emotions that she had at the time, frustration and anger for those that had taken advantage of her. Although I thought I knew a lot about my mom’s childhood, my grandma had told me her perspective and what she could’ve done instead.”(Week 4, #9)

#### 3.1.3. More Engagement

Another common theme that emerged from the weekly reflections was how the participants felt engaged with their grandparents through this experience. For example, participants said:

“This experience went smoothly… My grandma had stories prepared and was understandable throughout the discussion. During the conversation, I actually felt engaged and kept track of the story.”(Week 1, #1)

Another participant reflecting upon a week also mentioned how his grandparent, who had middle-stage dementia, was able to recognize him. “It was a good experience overall. My grandmother was able to recognize me and say my name, which made me feel so engaged”.(Week 3, #7)

One of the participants felt a heartwarming experience and was more engaged with their grandparent after the weekly conversation on the topic of “likes and dislikes”.

“This week felt a lot more open and engaged. I learned a lot about my grandmother from this talk on what she loves to do and things that don’t appeal to her very much. My grandmother is a very ‘in the moment’ person, she loves to travel immensely and takes interest on entertainment that feels real as if she is there. To me learning about these things about my grandmother is very heartwarming, since before I did not know much about her at all outside of her interest and love for religion. Outside of travel, she doesn’t hate anything and shows so much love without much resentment. Struggles/challenges with traveling she had told me was issues of income back then, after she moved to the U.S. there were times of taking multiple jobs to take care of her kids and put aside her hobbies that she loved to do. But from her eyes it was more important to get her kids into a private school/better education more than anything else.”(Week 4, #8)

### 3.2. Category 2: Challenging Experience

The participants, while they had a positive experience overall, faced some challenges during the study, as shown by the following themes.

#### 3.2.1. Over-Explaining Things/Going off Track from the Conversation

One of the first challenges that the participants faced was how their grandparents would tend to over-explain things. For example, one of the participants said, “My grandma would start over-explaining things because she did not think I understood. But this was easy, because I could just recount my notes and let her know I was on the same page as her.” (Week 1, #1)

The grandchild participants faced another challenge when the conversation veered off track, leading to discussions on unrelated topics, making it challenging for them to refocus and return to the original subject. One participant shared:

“The least helpful aspect of this session was the time tracking. My grandma loves to tell short stories that are loosely related to the topic and with this there were often time jumps. Yet, I felt bad stopping her train of thought. In the end, though, I think it is more important for me to note the actual content of the story than to get exact dates.”(Week 2, #1)

Another participant also stated something similar: “It was nice having such a broad collection of categories, but it was hard to stay on track with all the options and things she had wanted to share with me. I wasn’t against the sidetracks, but it was hard to get back to whatever person, place, or thing she was talking about.” (Week 4, #9)

#### 3.2.2. Language or Vocabulary Barriers

Quite a few participants talked about the challenges they faced due to vocabulary issues and language barriers. For example, one participant expressed:

“There were many vocabulary issues that I had trying to explain to my grandma that made it difficult for the conversation to continue, because she speaks Chinese. There are many different interpretations to the key words that were in the lesson.”(Week 1, #9)

Participants who were quite fluent in the language their grandparents spoke also faced difficulties when they had to translate technical words, as one participant expressed: “My Chinese ability is decent but struggles when we got into more technical words related to his field of research, etc. (Week 1, #5). Similarly, another participant said:

“I struggled a little bit sometimes to understand his answers to my questions because my Chinese vocabulary in “career” topics is not very strong. But I think I was able to understand main points, even if some of the details flew past me.”(Week 3, #4)

#### 3.2.3. Overly-Broad Topics

Another theme that emerged from the challenges that the participants faced was related to certain conversation topics or questions. Some participants thought some topics were challenging for them. For example, one participant thought the topic of week 4 (loves and hates or likes and dislikes) was too vague: “The topic can feel a bit vague at first. what do you mean love? What do you mean hate? a person, thing? etc.” (Week 4, #5). Other participants had concerns about the overly-broad topic. For example, participant #2 mentioned, “Because the topic was so broad, my grandmother seemed to struggle with honing in on one or two specific points in her life to focus on.” (Week 1, #2). Similarly, participant #9 expressed:

“It was nice having such a broad collection of topics, but it was hard to stay on track with all the options and things she had wanted to share with me. I wasn’t against the sidetracks, but it was hard to get back to whatever person, place, or thing she was talking about.”(Week 4, #9)

A few other participants thought the guiding questions were repetitive. Participant #12 reported: “I felt that the majority of the questions about a challenge was redundant.” (Weeks 2–6, #12)

### 3.3. Category 3: Strategy to Lead the Weekly Conversation

Though some challenges were experienced during their weekly conversation, the grandchild participants used different strategies to overcome the challenges and the weekly talk with their grandparents was smooth and fruitful. The participants also talked about the things that they found the most helpful during the weekly call process.

#### 3.3.1. Using Guiding Questions in the Manual

Different strategies worked for different participants, but one of the most common themes that emerged was about the icebreaker and guiding questions. Talking about the icebreaker question, one participant said, “The suggested questions/ice breakers really opened her to telling me her interests. Since I also want to travel/explore someday, we connected on the love for travel” (Week 4, #8). Other participants also expressed “The most helpful part of this session was the ice breaker questions” (Week 2, #1), and “The most beneficial was the icebreaker questions.” (Week 3, #6)

Talking about the guiding questions in the manual, one participant expressed, “The questions that were in the call manual was very helpful. It gave me ideas for other questions that I can ask and kept the conversation going.” (Week 1–4, #3). Talking about the importance and benefits of the guiding questions, a participant expressed:

“The questions are always a very good guide as to how I can further my understanding of the topic and what steps can be made to find more information. The questions were fairly simple, but regardless, they could garner more details for the situations.”(Week 2, #9)

Another participant also stated that talking about the guiding questions was helpful in directing the conversation with their grandmother:

“I really enjoyed the guiding questions. I really enjoyed asking her a set of questions regarding the individuals that she liked and asking her the same set of questions regarding the individuals that she disliked. When I asked her the questions, it helped me be direct in what I was asking her. Additionally, I never have asked her before to choose only one individual that she liked and disliked.”(Week 4, #6)

Reflecting on a particular question and how the prompts led to a deeper understanding of the experiences that their grandparent had, a participant said:

“The flexibility of what a stressful situation could mean was quite helpful, as when I had asked my grandparent generally what situation she had experienced in the past, it was hard for her to think of something initially. However, when giving her the suggestions that were prompted in the manual, it was easier for her to come up with a personal experience.”(Week 5, #9)

While most of the participants found the guiding questions to be helpful, one participant thought the guidance was too much, as participant #2 shared: “At some points, the questions from the discussion guide were a little too guided. When my grandmother didn’t answer exactly as laid out, I had to improvise some new questions to ask her along the way.” (Week 3, #2)

#### 3.3.2. Using Translators or Asking for Help from Mothers

Due to the language barrier some participants experienced, they turned to a translator app or asked their mothers to help them. One such participant stated, “I found it helpful to have my mom translate some words for me and use a translator to look up words I did not know.” (Week 1, #6)

Similarly, talking about how their mother was a huge help in conveying what they needed to their grandparents, one participant shared, “The most helpful part of this session was definitely having my mom translate a few pieces of the questions, but also coming up with examples to help my grandma realize the feeling of “stress”.” (Week 5, #1)

#### 3.3.3. Spending Time Together

Some of the participants found spending time together with their grandparents to be a really good strategy that helped the conversation. For example, one participant said, “I found that spending time with my grandmother over the phone was very helpful, especially with her current health situation.” (Week 1, #7)

Talking about spending time in person, one participant stated, “I think it was nice that we were able to do the talk in person, so he had time to gather his thoughts before he answered a few hours later.” (Week 4, #5). Again, talking about the benefits of meeting face to face, one participant said, “The most beneficial thing was we were face to face, so it was easier to talk.” (Week 1, #10)

#### 3.3.4. Taking Notes

While the participants found different strategies that worked for them when they were talking with their grandparents, some participants found note-taking was really helpful. For example, one participant expressed, “Taking notes helped me stay engaged throughout the session and also kept my facts in check.” (Week 1, #1)

## 4. Discussion

The present study developed an Intergenerational Grandparent–Grandchild Reminiscence Program to connect grandparents with their grandchildren in Asian American families in the DFW area. In this program, the grandparent engaged in weekly reminiscence-based conversations with their grandchildren over a span of 6 weeks. The primary aims were to enhance the emotional well-being of the grandparents and foster intergenerational bonding. While evaluating the first purpose necessitated relying on reports from the grandparents themselves, the present study solely utilized RADaR analyses to explore the self-reflections of the grandchildren after each weekly conversation, revealing promising indications of positive outcomes for the second purpose, promoting intergenerational connections. Overall, the findings of the study show that grandchildren had positive experiences during this intervention in that they had more connection with their grandparent, learned more about their grandparent’s past life experiences, and had more engagement. The grandchildren also reported challenging experiences, such as over-explaining things, language or vocabulary barriers, and overly-broad topics of conversation. However, the grandchildren came up with strategies to help them lead the discussion by using the guiding questions in the manual, using translators, spending time together with their grandparents, and taking notes during the conversation.

First, the grandchildren expressed positive experiences when talking with their grandparents. The participants reported that this reminiscence-based program was beneficial in helping them learn about their grandparents’ past life experiences and how they felt more emotionally and psychologically connected to them. This is consistent with the previous literature. For example, one study found that reminiscence programs were one of six elements that were found to be useful in the intergenerational dementia program [35]. Another study showed that reminiscence-based interventions had positive effects on the younger generations involved in the study as well after learning their life stories [36]. The literature also shows positive results in helping younger generations build relationships with older adults and positive shifts in their attitudes with respect to older adults [30]. Because of the benefits of more connections and engagement with grandparents indicated in this project, this reminiscence-based program can be useful in solving often-reported issues in Asian American families, such as intergenerational conflict and communication gaps [17].

Secondly, talking about the challenges faced during the program, one of the main challenges faced by the grandchildren was the language or vocabulary barriers between the grandparents and grandchildren. This is consistent with the literature, which shows in immigrant families, language barriers can play a crucial role in threatening the formation of the bond between the grandparents and the grandchildren [16]. The grandchildren participants in our study reported having difficulty in explaining topics to their grandparents, especially when they did not know a particular technical word in the native language of their grandparent. The participants also talked about challenges they faced while trying to talk about different topics that the grandchildren were familiar with, but the grandparents did not relate to, or vice versa. This might be partially due to the different acculturation levels between the two generations because accepting and adjusting to a new culture, as well as high acculturation have shown to be beneficial in sustaining close relationships among grandparents, the middle generation, and grandchildren [17].

Lastly, even though the grandchildren faced some challenges during the study, they were smart to come up with strategies and solutions to help smooth the weekly conversation. The grandchildren suggested that taking notes during the sessions helped them to keep the conversations going. Many of the participants fully took advantage of using the guiding questions and ice breakers listed in the calling manual. They mentioned that using these questions and prompts helped them keep the conversation going naturally and tackle the issues they faced with their grandparents over-explaining/going off track during conversations. Quite a few participants faced language barrier issues and they came up with the strategy of using translator apps to help them navigate the conversation, while some participants used help from their mothers as translators. These results indicate that an intergenerational program between grandparents and grandchildren in immigrant families is promising because both generations will use different strategies to overcome challenges, if possible.

### Study Limitations, Contributions, and Implications

Some limitations were noticed when interpreting the present study findings. First, due to the nature of the qualitative data, the sample was not differentiated by their demographic characteristics, such as gender, age, and education level, which may influence intragroup differences in their personal experience and perceptions. The weekly one-page reflection may also offer a limited depth of content as compared to regular in-depth interviews. Second, the intervention was conducted in the DFW area, representing one southern region of the U.S. It is thus hard to tell if the grandchildren in other states or regions had similar experiences or challenges. Lastly, the majority of the participants came from Chinese-American families, which cannot represent the large variations among Asian American families. Future studies may consider including more ethnic minority Asian American families to compare the different perspectives among ethnic Asian grandchildren.

Despite the study limitations, this study contributes to reminiscence therapy by confirming its great potential to be conducted by non-professional grandchildren volunteers for immigrant older adults who have cognitive impairments. This is important since reminiscence is typically implemented by trained professionals (e.g., social workers, nurses, psychologists). Though there has been growing interest in using trained volunteers due to staffing shortages and the costs associated with reminiscence programs, more research related to intergenerational reminiscence has been called for in the literature [37]. As part of efforts to utilize volunteers rather than professionals, this study fills in the literature gaps by connecting younger grandchildren volunteers with older grandparents. The results of this study also add to the existing positive findings of reminiscence programs with different study populations (immigrant older adults with dementia) and different approaches (intergenerational approach within a family).

Furthermore, the findings from the present study have important implications for practice and service provision for Asian American immigrant families. First, because of the promising results of the reminiscence program reflected by the grandchildren, practitioners or program managers can think about integrating reminiscence/life review into their regular programs that promote family cohesion and intergenerational programs for Asian American families. Second, when implementing an intergenerational reminiscence program, it will be helpful to prepare the working manual as the present study did. In the manual, the guiding questions on each topic can be narrowed down with less overlap. It will also be helpful to tell participants to not over-explain the conversation topics from the beginning. Moreover, given the common language barriers grandchildren face, programs such as volunteer programs or intergenerational programs have the potential to encourage and help the younger generation gain more language skills in the heritage language of their family and potentially make more connections with their grandparents.

## 5. Conclusions

Immigration, aging, and dementia often result in a triple jeopardy for Asian Americans, who face numerous challenges and physical, psychological, and mental health issues [8,9,10,11], as well as threatened intergenerational relationships [16,17]. To improve the emotional well-being of older adults and promote intergenerational bonding, an Intergenerational Grandparent-Grandchild Reminiscence Program was developed. By qualitatively analyzing the reflections of the grandchildren after weekly discussions, the themes identified by the RADaR analyses in the present study show that intergenerational reminiscence programs offer the potential to increase social connection and engagement between grandparents and grandchildren among Asian American families, and help grandchildren better understand older adults’ past lives before and after immigration. A specific focus on intergenerational reminiscence between grandparents and grandchildren may offer a valuable approach for increasing generational bonding and relationships, and thus, improve the psychological well-being of Asian immigrant older adults in the United States.

## Figures and Tables

**Table 1 behavsci-13-00733-t001:** Demographic of grandchildren participants (*n* = 12).

ID	Age	Education	Gender	Ethnicity	US-Born	Geographic Distance from GP	Aware of GP’s Memory Issues	SRH
#1	18	High school	Female	Chinese	Yes	Another state	Yes	Excellent
#2	26	Bachelor	Female	Chinese	Yes	Same state	Yes	Good
#3	20	Some college	Female	Chinese	Yes	Same state	No	Good
#4	20	Some college	Female	Chinese	Yes	Another state	No	Very good
#5	26	Bachelor	Male	Chinese	No	Another state	Yes	Very good
#6	21	Some college	Female	Vietnamese	Yes	Same state	Yes	Good
#7	25	Some college	Male	Indian	Yes	Another state	Yes	Good
#8	22	Bachelor	Male	Vietnamese	Yes	Same state	No	Good
#9	19	Some college	Female	Chinese	Yes	Same state	Yes	Good
#10	18	High school	Female	Chinese	Yes	Same city	No	Excellent
#11	21	Some college	Female	Chinese	Yes	Same house	Yes	Good
#12	18	High school	Male	Chinese	Yes	Another state	No	Very good

Note: GP = grandparent; SRH = self-rated health.

## Data Availability

The data presented in this study are available upon request from the corresponding author. The data are not publicly available due to their qualitative nature.

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
