# Peer review of "Learning about Grandparents’ Past Life: Reflections of Grandchildren in an Intergenerational Reminiscence Project for Asian American Families"

_behavsci, 2023, doi:10.3390/bs13090733_

Round 1
Reviewer 1 Report
The study deals with an important and up to date issue. The argument is clear and well-explained in the introduction. The manuscript is written in a distinct and coherent manner relevant for qualitative studies. The study is characterized by clearly presented design containing description of details of the data collection and of the data analysis. The findings well-structured. Implications and limitations are shown in the discussion section.
Author Response
RESPONSE: We would like to express our gratitude for the positive feedback received in this review. Your kind comments and encouragement are greatly appreciated.
Reviewer 2 Report
This study was a pilot study to report the qualitative outcome of an intergenerational reminiscence program in improving the well-being of older Asian Americans and generational bonding.
Introduction
The second paragraph: The vicious cycle of dementia and trauma was not clearly explained.
The third paragraph: Section 1.1 should be added here to fully explain the importance of the intergenerational reminiscence program in improving well-being in older Asian Americans.
The objective of this study was not clearly stated.
From the introduction, the authors stated the challenges facing by the fast growing Asian American older adults in US and the importance of improving their well-being.
But later in this study, it seemed to focus on the effect of the intergenerational reminiscence program in grandchildren more. This made me very confusing on the aim of this study.
The information in Section 1.2 should be presented in Materials and Methods.
Materials and Methods
Line 114, if this is a qualitative or both quantitative and qualitative study?
The inclusion criteria did not define the age group of older adults. It was very confusing in Line 124.
The demographic information of the recruited participants and Table 1 in Section 2.2 should be presented in Results. Table 1 reported the age of grandchild but not older adults. It was confusing if the subjects under investigation were grandparents (older adults) or grandchildren in this study.
Results
Category 2 and 3 presented here were different from that in abstract.
Discussion
Through this study, I cannot see the evidence in showing the intergenerational reminiscence program can improve the well-being of older Asian Americans.
Conclusion
It did not relate to the objective and introduction.
Author Response
This study was a pilot study to report the qualitative outcome of an intergenerational reminiscence program in improving the well-being of older Asian Americans and generational bonding.
Introduction
- The second paragraph: The vicious cycle of dementia and trauma was not clearly explained.
RESPONSE: We extend our gratitude to the reviewer for this valuable comment. In response, we have thoroughly revised the entire paragraph to enhance its clarity.
- The third paragraph: Section 1.1 should be added here to fully explain the importance of the intergenerational reminiscence program in improving well-being in older Asian Americans.
RESPONSE: We revised the 3rd paragraph as suggested by the reviewer.
- The objective of this study was not clearly stated.
RESPONSE: Thank the reviewer for this comment. We added a sentence about the objective of the research project under section 1.2. on page 5.
- From the introduction, the authors stated the challenges facing by the fast growing Asian American older adults in US and the importance of improving their well-being. But later in this study, it seemed to focus on the effect of the intergenerational reminiscence program in grandchildren more. This made me very confusing on the aim of this study.
RESPONSE: Initially, we highlighted the significant mental health challenges encountered by Asian American older adults and underscored the importance of investigating this topic for this particular population. Subsequently, we delved into the existing literature on evidence-based practices and interventions, such as reminiscence therapy and intergenerational connections, known to benefit older adults in general. Building upon these findings, we crafted our study to cater specifically to this population by amalgamating reminiscence and grandparent-grandchild connections. To enhance clarity, we have thoroughly revised the entire introduction section.
- The information in Section 1.2 should be presented in Materials and Methods.
RESPONSE: We moved the intervention process to 2.1. under “Process of Data Collection”, as suggested by the reviewer.
Materials and Methods
- Line 114, if this is a qualitative or both quantitative and qualitative study?
RESPONSE: Throughout the entire research project, we employed a combination of quantitative and qualitative study designs. However, in the context of this particular paper, our focus lies solely on presenting the qualitative aspect of the grandchildren's reflections. We have revised this paragraph to ensure its clarity for section 2.1.
- The inclusion criteria did not define the age group of older adults. It was very confusing in Line 124.
RESPONSE: We appreciate the reviewer’s comment, and in response, we have revised the inclusion criteria for clarity. For grandparents, the requirement now is that they must be aged 65 or above, while for the grandchildren, the eligible age range is 18 or above.
- The demographic information of the recruited participants and Table 1 in Section 2.2 should be presented in Results. Table 1 reported the age of grandchild but not older adults. It was confusing if the subjects under investigation were grandparents (older adults) or grandchildren in this study.
RESPONSE: Thank you for providing this valuable comment. After careful consideration, we have determined that it would be more suitable to present the demographic information of participants in Section 2.2. There are two main reasons for this decision. First, the present paper primarily focuses on a qualitative report of reflection pages, and as such, demographic information does not fall under the results/findings section, which is typically found in quantitative papers. Second, due to the numerous subheadings in the results/findings section, stemming from the categories and themes derived from content analyses, introducing an additional subheading for demographic information would disrupt the balance and coherence of the results/finding part. By locating the demographic information to Section 2.2, we aim to maintain clarity and organization in the presentation of our qualitative findings.
Results
- Category 2 and 3 presented here were different from that in abstract.
RESPONSE: Thank the reviewer for identifying this oversight. We have rectified the abstract section and ensured that the findings in both locations are now consistent.
Discussion
- Through this study, I cannot see the evidence in showing the intergenerational reminiscence program can improve the well-being of older Asian Americans.
RESPONSE: We sincerely appreciate the reviewer for offering this insightful comment. We wholeheartedly agree that one of our aims was indeed to improve the emotional well-being of the grandparents. However, evaluating this purpose necessarily rely on reports from the grandparents themselves, which will be presented in another manuscript in the near future. Consequently, we have incorporated this clarification into the first paragraph of the discussion section on page 18.
Conclusion
- It did not relate to the objective and introduction.
RESPONSE: Thank the reviewer this comment. We revised the conclusion part to make sure it relates to the objective and introduction part.
Reviewer 3 Report
I believe it is a good attempt to analyze the reminiscence that grandchildren may have by experiencing indirectly the past lives of their grandparents. If only a few things related to manuscript writing are supplemented, it will be evaluated that can be published in journals. What I think needs to be improved is:
1. Please, do not number in the abstract.
2. In the introduction, please present the necessity of this study in more detail, and describe the rationale that can achieve the purpose of the study and draw a conclusion with this research design in a more convincing way.
3. Didn't you collect more qualitative data than the one-page reports presented by your grandchildren? I think it would be difficult to include deep content in a one-page report.
4. You have explained the qualitative analysis process well enough. You have also explained the process of the Intergenerational Reminiscence Project well, but it would be more helpful for the reader to be a bit more detail.
5. There are relatively little discussions about the results of this study. Especially, provide more specific clinical implications.
Author Response
I believe it is a good attempt to analyze the reminiscence that grandchildren may have by experiencing indirectly the past lives of their grandparents. If only a few things related to manuscript writing are supplemented, it will be evaluated that can be published in journals. What I think needs to be improved is:
- Please, do not number in the abstract.
RESPONSE: We removed the numbers in the abstract as suggested.
- In the introduction, please present the necessity of this study in more detail, and describe the rationale that can achieve the purpose of the study and draw a conclusion with this research design in a more convincing way.
RESPONSE: Thank the reviewer for this feedback. We have thoroughly revised the introduction to ensure smoother transitions throughout. Initially, we introduced the pressing mental health issues that Asian American older adults face, highlighting the significance of studying this topic for this particular population. Subsequently, we delved into the literature, exploring evidence-based practices and interventions beneficial for older adults in general, including reminiscence and intergenerational connections.
Building upon this existing literature, we tailored our study to address the needs of this specific population by integrating reminiscence therapy with grandparent-grandchild connections. In conclusion, we provided a concise overview of our research design and heeded the suggestion of another reviewer by moving the detailed design to the "method" section.
- Didn't you collect more qualitative data than the one-page reports presented by your grandchildren? I think it would be difficult to include deep content in a one-page report.
RESPONSE: Following each week's conversation, each grandchild participant submitted a one-page reflection page, resulting in a total of six reflections for each participant. It's worth noting that some participants went into great detail and exceeded one page in their reflection submissions. We had made revisions to improve clarity in this regard on pages 6-7, and we also acknowledged this particular limitation in the discussion section on page 20.
- You have explained the qualitative analysis process well enough. You have also explained the process of the Intergenerational Reminiscence Project well, but it would be more helpful for the reader to be a bit more detail.
RESPONSE: Thank the reviewer for this comment. We have added relative information about the Intergenerational Grandparent-grandchild Reminiscence program in the introduction part and method part, as highlighted in yellow color.
- There are relatively little discussions about the results of this study. Especially, provide more specific clinical implications.
RESPONSE: In the discussion section, we dedicated three distinct paragraphs to reexamine the themes derived from three categories, corresponding to the results/findings section. Additionally, on page 21, just before the "conclusion" section, we included a separate paragraph focusing on the practical implications and potential services stemming from our study.
Round 2
Reviewer 2 Report
Thank you for addressing my previous concerns.